# Research on Multi-Objective Optimization on Explosion-Suppression Structure-Nonmetallic Spherical Spacers

**Minjie Liu [1], Yangyang Yu [1,2,*], Junhong Zhang [1,2,*], Dan Wang [1], Xueling Zhang [1] and Meng Yan [3]**

[1] Tianjin Renai College, Tianjin 301636, China; 093015@tjrac.edu.cn (M.L.); wangdan1988@tjrac.edu.cn (D.W.); 133016@tjrac.edu.cn (X.Z.)

[2] State Key Laboratory of Engine, Tianjin University, Tianjin 300072, China

[3] Development Services Center of Agriculture, Jinghai District of Tianjin, Tianjin 301600, China; 211028@tjrac.edu.cn

\* Correspondence: yuyangyang@tju.edu.cn (Y.Y.); zhangjh@tju.edu.cn (J.Z.); Tel.: +86-139-2027-3705 (Y.Y.); +86-136-0205-7962 (J.Z.)

**Abstract:** Intense burning phenomena (fire disasters) need to be prevented in the combustible gas utilization and transportation processes to ensure industrial safety. Nonmetallic spherical spacers (NSSs) have been investigated and applied in lots of explosive atmospheres to prevent explosion execution in a confined space. In this work, a novel fuzzy-based analytic hierarchy process (FAHP) is developed to take into account the uncertainty in decision-making and effectively solve the problem of factor weight allocation in multi-objective optimization. Optimal Latin Hypercube Design (Opt LHD), Chebyshev Orthogonal Polynomials (COP), and Adaptive Simulated Annealing (ASA) were combined. A multi-objective optimization method is proposed for the structural parameter optimization problem on NSSs in order to achieve conflicting multiple-objective optimization of low displacement rate and minimal deformation. That is to say, the small volume (low displacement rate) and high explosion-suppression performance (minimal deformation) of NSSs were optimized simultaneously. The results show that, compared with the original NSS model's deformation (2.85 mm) and displacement rate (3.63%), the optimized NSSs with weight allocation had optimized the deformation by 12.98% and displacement rate by 6.1%. Compared with the optimized design model of NSSs without weight allocation with a deformation of 2.75 mm and a displacement rate of 3.48%, the deformation has been optimized by 9.82%, and the displacement rate has been optimized by 2.0%. It was verified that the proposed method is effective. At the same time, it was verified that the suppression effect of NSSs can be enhanced by changing the shape of the NSS spacer reasonably by experimental verification.

**Keywords:** fire prevention; novel fuzzy analytic hierarchy process; multi-objective optimization; weight allocation; nonmetallic spherical spacers

## 1. Introduction

Flame-vortex-induced flame propagation acceleration [1,2]. Under suitable conditions, uncontrollable autoignition occurs on trapped and compressed end-gas, resulting in DDT. The perforated plate is usually set to reduce the impact of fuel inertia on the fuel vessel. There is a general perception that the flame propagation rate is enhanced when passing through perforated plates [3–6]. The flame passing through perforated plates will induce higher flame velocity and further lead to overpressure and end-gas autoignition phenomena [3], which may exacerbate the explosion process. The flame- perforated plate interaction will result in higher flame velocity and further lead to overpressure and end-gas autoignition phenomena [7]. The damage caused by overpressure and end-gas autoignition is more harmful than that caused by the ensuing fires [8]. However, both intense burning phenomena (fire disasters) need to be inhibited in the utilization and transportation processes to ensure industrial safety.

Based on the flame blocking effect and heat conduction effect [9,10], explosion-suppression structures such as nonmetallic spherical spacers (NSSs) have been investigated and applied in lots of explosive atmospheres to prevent explosion execution in a confined space. Yang [11] analyzed the effects of propane explosion overpressure on NSSs in a long cylindrical tube. NSSs showed excellent suppression characteristics. Lu [12] verified the explosion-suppression performance of NSSs through shock tube tests, equivalent static explosion tests, and deflagration bomb tests. Zhao [13] set up a mesoscale circular tube confined space experimental bench and confirmed that NSSs reduced the overpressure of gasoline-air mixture explosions, weakened turbulence development, and accelerated the oscillating strengthening process. Flame intensity was tested in a closed space. Although flame propagation was not completely prevented, NSSs shortened the flame duration and reduced the flame intensity. Our work [14] researched the energy attenuation behavior of propane-air premixed combustion on account of NSSs in a confined space with two perforated plates. NSSs show a good energy attenuation effect on propane normal combustion, jet autoignition, and deflagration evolution behavior; however, during the evolution of weak detonation, the attenuation effect is relatively weak. But Song [15] also discovered that the overpressure of hydrogen explosion under the influence of NSSs increased gradually; NSSs promoted hydrogen explosion and formed detonation. As a result, high-intensity flame evolution was not suppressed by NSSs. But inhibiting effects can be enhanced by structural optimization of NSSs, and the influences of NSS construction under high-intensity combustion modes need to be in-depth researched.

Combustible oil and gas explosions were suppressed by filling NSSs in confined spaces, such as fuel tanks, gas tanks, and oil and gas pipelines. Combustion was easily induced on account of improper operation. The risk of deflagration or detonation was increased in enclosed spaces with double-perforated plates on account of accelerating flames [16]. Under the impact of flame acceleration, the spacer structure of NSSs undergoes deformation, absorbs energy, and attenuates energy. Flames with higher combustion intensity can cause significant deformation, directly affecting structural stability and explosion suppression performance. On the other hand, our work typically considers lightweight factors, focusing on mass and strength, when optimizing the structural parameters of NSSs [17]. The optimization problems of the NSS structure were complex, with multiple variables and nonlinear relationships. Engineering goals were influenced by many factors that interact with each other; it was difficult to define the degree of their impact on the goal using an "either/or" approach. Especially when multiple conflicting objectives were involved, the contribution of each factor needed to be quantitatively represented. In multi-objective optimization methods, factor weight assignment was usually determined by engineering experience, which was subject to significant errors due to human subjectivity and uncertainty. Therefore, multi-criteria and multi-factor decision-making methods that are suitable for solving complex problems, as well as more accurate and efficient multi-objective optimization research, have important theoretical value and engineering significance. The Analytic Hierarchy Process (AHP) is a widely used multi-criteria decision-making method that determines the weight of each criterion and the priority of each alternative solution in a structured manner through pairwise comparisons. This method has been widely used in many fields [18–22]. On the contrary, some limits have been found for the traditional AHP approach [23]: (1) the consistency of the judgment matrix is very costly to achieve; (2) preferences mismatch the objective priority; and (3) the uncertainty of subjective evaluation reduces the accuracy. To overcome the above shortages, fuzzy-based AHP (FAHP) was developed to take into account the uncertainty in decision-making. A fuzzy scale is used to express the preference or relative importance of the target property instead of adopting an exact value from a pairwise comparison. In addition, the fuzzy consistency matrix [24] used in the FAHP can be determined by membership functions, so the consistency of the judgment matrix is met automatically [25]. The traditional weighting approach was improved by the FAHP.

Research on multi-objective decision-making mainly focuses on the application of solving problems in different fields. The research methods mainly focus on the combination of various methods such as AHP and the entropy method, TOPSIS, fuzzy set theory, and gray correlation. Ashish [26] proposed a hybrid multi-objective decision-making model based on AHP, fuzzy set theory, and the goal programming method. The model used the goal programming method, transforming multiple objectives into a single objective function in order to find a solution. It is used to select compromise solutions for multiple objectives in post-disaster recovery projects. Guo [27] combined AHP and TOPSIS based on fuzzy rough numbers and proposed a fuzzy rough number-enhanced design concept evaluation group decision-making framework. Through research, it has been shown that the fuzzy rough number method has superiority in dealing with the uncertainty and subjectivity of design concept evaluation in group decision-making environments. Lishu et al. [28] proposed a multi-objective decision-making method for processing technology schemes. The method is based on AHP to determine the combination weights, and the importance of indicators is determined by considering the correlation between subjective and objective weights using the CRITIC method. Then, the TOPSIS method was used for similarity ranking, setting positive and negative reference sequences, and using gray correlation and Euclidean distance to rank each experimental group (processing parameters) to determine the best process parameters, achieving multi-objective coordinated optimization. However, the low displacement rate and high explosion-suppression performance of NSSs were optimized simultaneously on structural parameters; allocating weights has problems such as poor objectivity in determining indicator weights and difficulty in verifying the consistency of judgment matrices. And the allocation weights of multiple objectives need to be developed.

In this work, a novel fuzzy-based analytic hierarchy process (FAHP) is developed to take into account the uncertainty in decision-making and effectively solve the problem of factor weight allocation in multi-objective optimization. Optimal Latin Hypercube Design (Opt LHD), Chebyshev Orthogonal Polynomials (COP), and Adaptive Simulated Annealing (ASA) were combined. A multi-objective optimization method is proposed for the structural parameter optimization problem on NSSs in order to achieve conflicting multiple-objective optimization of low displacement rate and minimal deformation. That is to say, the small volume (low displacement rate) and high explosion-suppression performance (minimal deformation) of NSSs were optimized simultaneously. At the same time, it was verified that the explosion-suppression effect of NSSs can be enhanced by changing the shape of the NSS spacer reasonably by experimental verification.

## 2. Analytical Method

### 2.1. Multi-Factor Analysis of Variance

Many changes in phenomena are the result of multiple factors. The conditional error and random error were decomposed from the total variation by a multi-factor analysis of variance (MANOVA) involving the factorability of the mathematical model, and then the main factors affecting the test results from multiple factors were found.

The mathematical model of MANOVA is established, which obeys the normal distribution $X_{ijk} \sim (\mu, \sigma^2)$:

$$x_{ijk\cdots} = \mu + a_i + b_j + \cdots + (ab)_{ij} + \cdots + \varepsilon_{ijk\cdots} \ (i = 1, 2, \cdots, m; j = 1, 2, \cdots, r; k = 1, 2, \cdots, l; \cdots) \tag{1}$$

where $a_i$ and $b_j$ are the main effects, $\varepsilon_{ijk}$ is the independent random error, $(ab)_{ij}$ is the interaction effect, $\mu$ is the average value, $\sigma$ is the variance.

The variance of each effect is decomposed by the mutation separator of $x_{ijk\cdots}$. The mean square value of each effect was acquired by dividing the variance of each effect by the degrees of freedom. The F-value for each effect was acquired by the mean square of each effect divided by the mean square error. The probability $P$ of $\geq F$ was acquired by F-distribution. The *p*-value is used to determine whether the effect term has a significant influence on the dependent variable. If $p > 0.05$, it indicates that the effect term has no

significant effect on the dependent variable. If $p < 0.05$, it indicates that the effect term has a significant influence on the dependent variable. According to the F-value, the degree of influence of each effect term can be judged on the dependent variable.

In our work, this method is applied to address the multi-structure contribution problem on NSSs. The contributions of the NSS structural parameter variables were determined by MANOVA on deformation and displacement rate, respectively.

## 2.2. The Novel Fuzzy Analytic Hierarchy Process (FAHP) Method

The optimization objective was directly affected by the weight allocation of factors in the optimization process. A fuzzy judgment matrix that is more in line with the actual situation was constructed by FAHP. The "reversed order" issue in AHP can be effectively solved due to the "additivity" of the fuzzy judgment matrix. In addition, the fuzzy judgment matrix can be converted into a fuzzy consistency matrix through membership relations, thus effectively avoiding the problem of consistency in the AHP judgment matrix.

The detailed process of the FAHP can be described as follows:

Step 1: Establish the fuzzy judgment matrix **A** from pairwise comparisons. Determine the fuzzy judgment matrix $\mathbf{A} = [a_{ij}]_{n \times n}$ according to the relative importance of factors in the same layer. The elements in matrix A are in the range of 0.1 to 0.9, where 0.5 represents that the two factors are equally important.

Step 2: Calculate the fuzzy consistency matrix **R** and the reciprocal matrix **M**. Transform $\mathbf{A} = [a_{ij}]_{n \times n}$ into $R = [r_{ij}]_{n \times n}$. The calculation can be performed using the following equation:

$$r_{ij} = \frac{1}{2n} \left( \sum_{k=1}^{n} a_{ik} - a_{jk} \right) + 0.5 \tag{2}$$

Then, the matrix **R** can be converted into the reciprocal fuzzy consistency matrix **M** through $m_{ij} = r_{ij}/r_{ji}$.

Step 3: Calculate the initial weighting vector $\omega_0$. The initial weighting vector $\mathbf{\omega_0} = [\omega_1 \ \omega_2 \cdots \omega_n]$ can be obtained by the least-squares method:

$$\begin{cases} \omega_0 = \frac{1}{n} - \frac{1}{2\partial} + \frac{1}{n\partial} \sum_{j=1}^{n} r_{ij} \\ \partial = \frac{n-1}{2} \end{cases} \tag{3}$$

Step 4: Calculate the final weighting vector W:
(1) Initialize $W_0 = [v_{0,1} \ v_{0,2} \cdots v_{0,n}]$ by $\omega_0$;
(2) For $k = 1, 2, 3, \ldots, n$,

$$W_{k+1} = \boldsymbol{M} W_k. \tag{4}$$

(3) Check the convergence by using the following equation:

$$||W_{k+1}||_\infty - ||W_{k+1}||_\infty \le \varepsilon \tag{5}$$

(4) If convergence is met, output the final weighting vector as follows:

$$W_{k+1} = \left[ \frac{W_{k+1,1}}{\sum\limits_{i=1}^{n} W_{k+1,i}} \ \frac{W_{k+1,2}}{\sum\limits_{i=1}^{n} W_{k+1,i}} \ \frac{W_{k+1,3}}{\sum\limits_{i=1}^{n} W_{k+1,i}} \cdots \frac{W_{k+1,n}}{\sum\limits_{i=1}^{n} W_{k+1,i}} \right] \tag{6}$$

Otherwise, jump back to Step 2.

Step 5. Calculate the combined weight, $W^k$.

Suppose the weight vector of $k - 1$ layer elements towards the overall goal is $W^{k-1}$, and the weight of the $k$ layer elements towards the $k - 1$ layer elements $C_j^{k-1}$ is $P_j^k$. The specific description is as follows:

$$W^{k-1} = (w_1^{k-1}, w_2^{k-1}, \ldots, w_{n_{k-1}}^{k-1})^T \tag{7}$$

$$P_j^k = \left(p_{1j}^k, p_{2j}^k, \ldots, p_{n_k j}^k\right)^T, j = 1, 2, \ldots, n_{k-1} \tag{8}$$

$$P^k = [P_1^k, P_2^k, \ldots, P_{n_k}^k] \tag{9}$$

Matrix $P^k$ is a matrix of dimension $n_k \times n_{k-1}$. The weight matrix of $k$ layer elements to the total target is $W_k$.

$$W^k = P^k \cdot W^{k-1} = \left(w_1^k, w_2^k, \ldots, w_{n_k}^k\right)^T \tag{10}$$

Step 6. A good element weight distribution can be obtained for impact elements of a single-objective optimization problem by the FAHP method. However, in practical engineering, multi-objective optimization problems are often urgent to solve. Multiple elements interact with each other for multi-objective optimization problems. The element weight distribution needs to consider multiple optimization objectives. Such problems were solved by improving the FAHP method; thus, a comprehensive optimization result can be achieved for multiple objectives. The improvement method is as follows:

According to Formula (10), calculate the weight matrix $(w_1^{ki}, w_2^{ki}, \ldots, w_j^{ki}, \ldots, w_{n_k}^{ki})^T$ of each individual objective. Assume that the multi-objective is a total target $W_t = \{W^{k1}, W^{k2}, \ldots, W^{kn}\}$, and each individual objective $W^{ki}$ is an element. According to the FAHP method, the weight matrix of the overall target $\{W_{k_t}^1, W_{k_t}^2, \ldots, W_{k_t}^n\}$ was obtained. Then, based on the weight matrices of $W^{ki}$ and $W_t$, the weight matrix of $W^k$ was reallocated to obtain a comprehensive multi-objective weight allocation matrix:

$$W^{k\prime} = \left(w_1^{k\prime}, w_2^{k\prime}, \ldots, w_j^{k\prime}, \ldots, w_{n_k}^{k\prime}\right)^T \tag{11}$$

where $w_j^{k\prime} = W_t \cdot \left(w_j^{k1}, w_j^{k2}, \ldots, w_j^{kn}\right)$.

Individual objectives were treated as elements, and a more realistic fuzzy judgment matrix that better reflects the actual working conditions was reconstructed. Then, through membership relationship transformation, a fuzzy consistent matrix was obtained, and in order to calculate the multi-objective weight allocation matrix, the weights were reallocated. The weights of the elements were allocated by calculating $W^{k\prime}$. The problem of factor weight allocation was solved in multi-objective optimization. Therefore, in this work, the problem of FAHP's inability to allocate factor weights in multi-objective optimization was solved by improving the FAHP method.

This method was developed to take into account the uncertainty in decision-making and effectively solve the problem of factor weight allocation in multi-objective optimization. In our work, taking into account both the deformation and displacement rate objectives, the weights of $R_1$, $R_2$, $R_3$, and $R_4$ of NSSs were calculated by Equation (11).

## 3. Multiple-Objective Optimization Method

The multi-objective optimization method of structure optimization is developed by MANOVA and the improved FAHP method (addressing conflicting multiple objective weight allocations), integrating Opt LHD, COP, and ASA. This method is applied to address the multi-objective structure optimization problem on NSSs in order to achieve the multiple objectives of low displacement rates and minimal deformation.

### 3.1. Multiple Objective Optimizations of NSS Structure

Combustible oil and gas explosions were suppressed by filling NSSs in confined spaces, such as fuel tanks, gas tanks, and oil and gas pipelines. Combustion was easily induced on account of improper operation. Large fuel tanks, gas tanks, and pipelines usually have built-in double-perforated plates, which prevent the shock effect. However, the risk of deflagration or detonation was increased in enclosed spaces with double-perforated plates on account of the accelerating flame. Under the impact of flame acceleration, the spacers of NSSs undergo deformation. Flames with high-intensity combustion can cause significant deformation, directly affecting structural stability and explosion suppression performance.

The minimal deformation indicates the outstanding fire resistance of NSSs and their explosion suppression performance. Our research [17] found that explosion suppression performance is proportional to spacer strength (affecting deformation). In addition, NSSs should occupy a small actual volume in fuel tanks or gas tanks. A lightweight factor was typically considered in structural design; a relatively large percentage of space while having a small actual volume requirement was occupied by the NSS structure. That is to say, it indicates a low displacement rate. It is an important criterion in the application of NSSs. Both a low displacement rate and minimal deformation were in conflict with each other and needed to be achieved simultaneously. Thus, it is a difficult question. In this work, considering both minimal deformation and a low displacement rate under accelerated flame impact, the structure of NSSs was optimized. The goal is to enhance the explosion suppression performance of NSSs while reducing the displacement rate. The optimization process is illustrated in Figure 1. First, the strength simulation and volume calculation were conducted on the initial model of NSSs. Deformation and displacement rates were produced under different conditions. Secondly, the deformation and displacement rate prediction models of NSSs were obtained by the Opt LHD and COP. The structural parameter variables were optimized for the deformation and displacement rates of NSSs by the ASA optimization algorithm. Then, the contributions and weights of the structural parameter variables were determined by MANOVA and improved FAHP. Once again, the structural parameter variables were optimized by the COP and ASA optimization algorithms. The results of two calculations were compared. Finally, the results were verified by experiment.

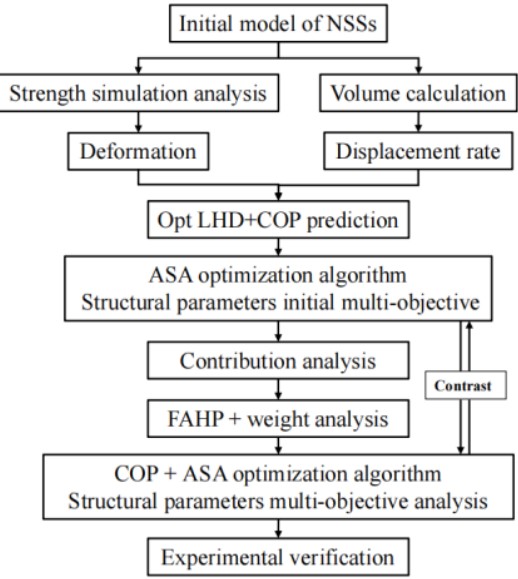

**Figure 1.** Procedure of the structure parameter optimization.

### 3.2. The Optimization Goals of NSSs

The spacers of NSSs undergo significant deformation and can even be severely damaged under the impact of a quasi-detonation flame [14]. Spacers in NSSs resisted deformation and absorbed energy, thus suppressing flame evolution [29]. According to reference [29], the overpressures had been tested under quasi-detonation conditions in a confined space where NSSs are placed. With a pressure of 4.98 MPa as the boundary condition on the flame contact surface in the finite element model, the deformation results were obtained by simulation analysis. The maximum deformation of spacers under the impact of the accelerated flame reaching quasi-detonation is shown in Figure 2, with the position marked by a white circle indicating the highest deformation, measuring 2.85 mm.

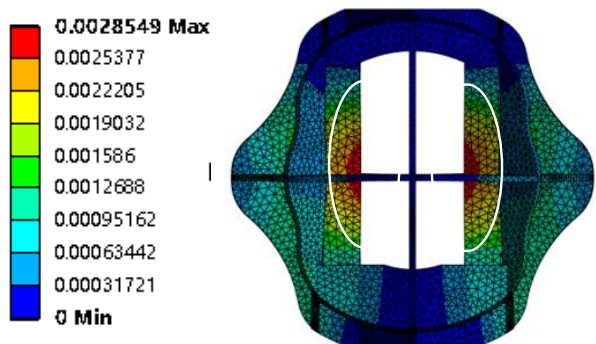

**Figure 2.** Deformation analysis of NSSs.

According to the calculation method of displacement rate specified in the Chinese standard JT/T 1046-2016, «Technical Requirements for Explosion-proof Safety of Road Transport Vehicle Fuel Tanks and Liquid Fuel Transportation Tanks» [30], the displacement rate $\Psi$ is as follows:

$$\Psi = \frac{V_1 - 1800}{1800} \times 100\% \tag{12}$$

where $V_1$ is the volume of the contacting medium after immersion in explosion-suppression material (mL).

The optimized-designed NSSs are filled in the 2000 mL volumetric measuring cylinder model, reaching a position of 1800 mL. The filling method is shown in Figure 3, with a total of 84 NSSs being filled. The volume of a single NSS is $V = 776.9$ mm$^3$. According to Formula (12), the displacement rate of the designed NSSs is calculated to be 3.63%.

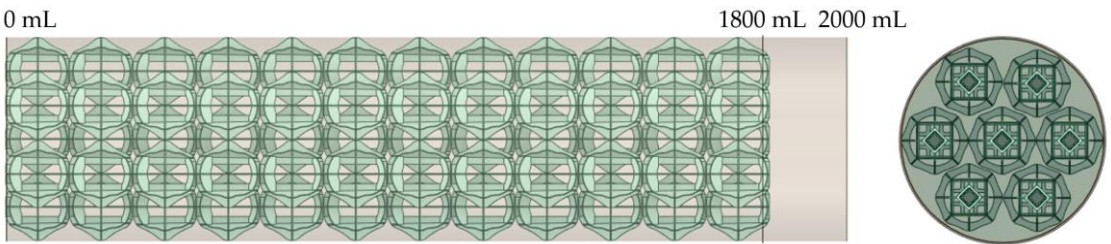

**Figure 3.** Distribution of NSSs in measuring cylinder.

As shown in Figure 4, the spacer structural parameters $R_1$, $R_2$, $R_3$, and $R_4$ of NSSs do not affect the overall space. However, they have a significant impact on the deformation of the structural components in the strength analysis. Therefore, these parameters are chosen as optimization parameters for multi-objective optimization design, aiming to achieve a low displacement rate and minimal deformation simultaneously.

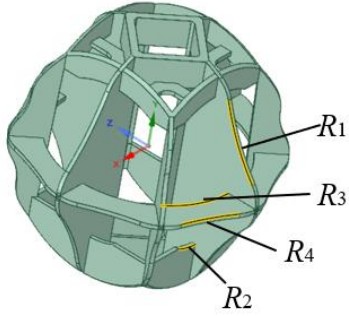

**Figure 4.** Four structure parameters of NSSs.

### 3.3. Sampling Structural Parameters

The range intervals of structural parameters $R_1$, $R_2$, $R_3$, and $R_4$ are shown in Table 1. A total of 20 samples were taken by the Opt LHD method, and the displacement rate and deformation were calculated through Equation (12) and strength simulation analysis, as shown in Table 2.

**Table 1.** Variable range of the structure parameters.

| Parameters | $R_1$ (mm) | $R_2$ (mm) | $R_3$ (mm) | $R_4$ (mm) |
|---|---|---|---|---|
| Variable scope | 2–30 | 6.8–9.7 | 9.9–12.5 | 6–50 |

**Table 2.** Samples of the structure parameters.

| No. | $R_1$ (mm) | $R_2$ (mm) | $R_3$ (mm) | $R_4$ (mm) | Deformation (mm) | Displacement Rate (%) |
|---|---|---|---|---|---|---|
| 1 | 21.16 | 7.56 | 10.58 | 50.00 | 2.93 | 3.62 |
| 2 | 16.74 | 8.17 | 11.13 | 29.16 | 3.54 | 3.49 |
| 3 | 25.58 | 7.72 | 12.23 | 43.05 | 4.48 | 3.38 |
| 4 | 9.37 | 7.11 | 10.99 | 10.63 | 3.29 | 3.51 |
| 5 | 4.95 | 8.33 | 11.54 | 47.68 | 4.00 | 3.33 |
| 6 | 24.11 | 9.39 | 11.27 | 45.37 | 3.37 | 3.37 |
| 7 | 13.79 | 8.94 | 11.95 | 8.32 | 4.70 | 3.26 |
| 8 | 28.53 | 6.95 | 10.86 | 24.53 | 3.08 | 3.61 |
| 9 | 27.05 | 8.48 | 9.90 | 31.47 | 2.41 | 3.64 |
| 10 | 12.32 | 9.24 | 10.17 | 40.74 | 2.70 | 3.50 |
| 11 | 22.63 | 8.02 | 10.45 | 6.00 | 2.95 | 3.57 |
| 12 | 3.47 | 9.55 | 11.41 | 26.84 | 4.02 | 3.19 |
| 13 | 2.00 | 7.87 | 12.09 | 22.21 | 4.50 | 3.18 |
| 14 | 6.42 | 8.63 | 10.31 | 12.95 | 2.86 | 3.48 |
| 15 | 30.00 | 8.78 | 11.82 | 19.89 | 4.00 | 3.42 |
| 16 | 7.89 | 7.41 | 10.04 | 33.79 | 2.50 | 3.63 |
| 17 | 19.68 | 9.70 | 10.72 | 17.58 | 3.26 | 3.38 |
| 18 | 10.84 | 6.80 | 11.68 | 38.42 | 3.71 | 3.45 |
| 19 | 18.21 | 7.26 | 12.36 | 15.26 | 4.52 | 3.35 |
| 20 | 15.26 | 9.09 | 12.50 | 36.11 | 5.27 | 3.19 |

### 3.4. Deformation and Displacement Rate Prediction

According to the aforementioned samples, in this work, predictive models of deformation and displacement rates were conducted by three fitting methods, namely COP, RBF, and RSM. The predicted results of these three different methods were compared with the numerical experimental results, as shown in Figures 5 and 6. The comparison of the results obtained from the three prediction systems is presented in Table 3. From the comparison in Table 3 and the accompanying figures, it can be observed that the COP prediction results are most consistent with the numerical experimental results compared with the RSM and RBF methods. This clearly indicates that the COP prediction system exhibits the highest prediction accuracy among these three methods.

The fitting accuracy of the established surrogate model can be evaluated by R-squared ($R^2$). The higher the $R^2$, the better the model. The formula for $R^2$ is as follows:

$$R_1{}^2 = 1 - \frac{\sum_{i=1}^{n}(X_i - Y_i)^2}{\sum_{i=1}^{n}(Y_i - L_i)^2} \tag{13}$$

The R-squared values of the three prediction models are as follows: For the COP method, the R-squared value of deformation is 0.98552, and that of displacement rate is 0.9704. For the RBF method, the R-squared value of deformation is 0.97514, and that of displacement rate is 0.96584. For the RSM method, the R-squared value of deformation is 0.95093, and that of displacement rate is 0.96187. It can be observed that the COP

prediction results are most consistent with the numerical experimental results. Therefore, the subsequent analysis and optimization will be conducted by the COP prediction system.

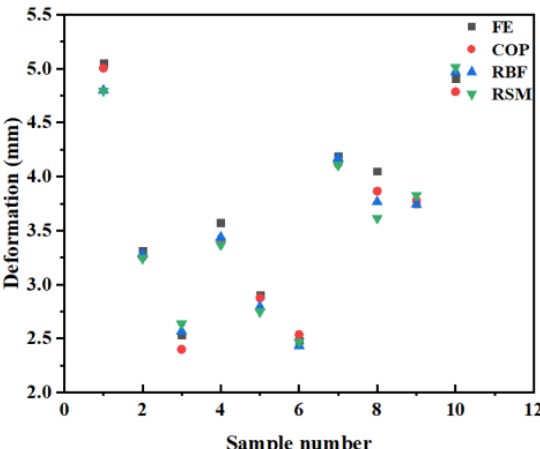

**Figure 5.** Comparisons of the predicted deformation.

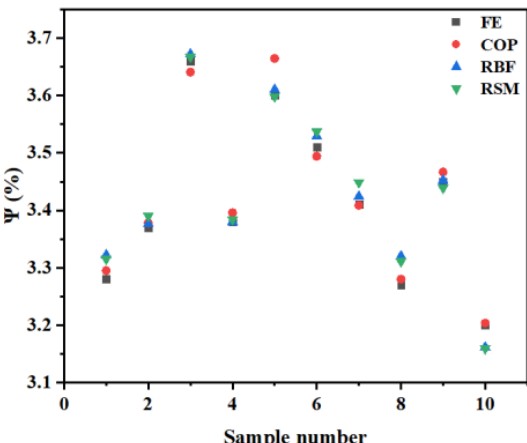

**Figure 6.** Comparisons of the predicted Ψ.

**Table 3.** Comparison between three prediction systems and FE analysis.

| No. | $R_1$ (mm) | $R_2$ (mm) | $R_3$ (mm) | $R_4$ (mm) | Deformation (mm) | | | | Displacement Rate (%) | | | |
|---|---|---|---|---|---|---|---|---|---|---|---|---|
| | | | | | FE | COP | RBF | RMS | FE | COP | RBF | RMS |
| 1 | 26.89 | 8.41 | 12.5 | 40.22 | 5.05 | 5.004 | 4.799 | 4.794 | 3.28 | 3.295 | 3.322 | 3.315 |
| 2 | 20.67 | 9.7 | 10.77 | 15.78 | 3.31 | 3.253 | 3.280 | 3.242 | 3.37 | 3.378 | 3.376 | 3.390 |
| 3 | 14.44 | 6.8 | 10.19 | 20.67 | 2.53 | 2.400 | 2.566 | 2.634 | 3.66 | 3.641 | 3.672 | 3.667 |
| 4 | 17.56 | 9.38 | 11.06 | 50 | 3.57 | 3.417 | 3.437 | 3.366 | 3.38 | 3.396 | 3.380 | 3.384 |
| 5 | 30 | 8.09 | 10.48 | 35.33 | 2.90 | 2.874 | 2.793 | 2.747 | 3.60 | 3.664 | 3.610 | 3.598 |
| 6 | 5.11 | 8.73 | 9.9 | 30.44 | 2.48 | 2.537 | 2.433 | 2.468 | 3.51 | 3.494 | 3.529 | 3.537 |
| 7 | 23.78 | 7.44 | 11.92 | 10.89 | 4.19 | 4.117 | 4.172 | 4.105 | 3.41 | 3.408 | 3.424 | 3.448 |
| 8 | 2 | 7.77 | 11.34 | 6 | 4.05 | 3.864 | 3.764 | 3.615 | 3.27 | 3.28 | 3.320 | 3.311 |
| 9 | 11.33 | 7.12 | 11.63 | 45.11 | 3.75 | 3.778 | 3.744 | 3.826 | 3.45 | 3.467 | 3.452 | 3.439 |
| 10 | 8.22 | 9.06 | 12.21 | 25.56 | 4.91 | 4.786 | 4.968 | 5.013 | 3.20 | 3.204 | 3.161 | 3.159 |

### 3.5. Multi-Objective Optimization of NSS Structural Parameters

Tables 2 and 3 show that the low displacement rate and minimal deformation are conflicting. Both conflicting multiple objectives optimization problems need to be solved, that is to say, the small volume (low displacement rate) and high explosion-suppression

performance (minimal deformation) of NSSs were optimized simultaneously. The multi-objective optimization problem of NSS deformation and displacement rate on structural parameters was calculated by a combination of the COP prediction model and the ASA optimization algorithm.

Objective: Find $y = f(R_1, R_2, R_3, R_4)$. Minimize: Deformation ($y_1$), displacement rate ($y_2$). Constraint conditions: as shown in Table 1. Here, $R_i$ ($i$ = 1, 2, 3, 4) represents the structural parameter variables.

The minimum and maximum values of the design variables in the constraint conditions are determined according to material characteristics and engineering experience. The deformation and displacement rates are determined by the COP-fitted model during the optimization iteration process. The maximum number of iterations was set to 200. As shown in Table 4, the optimal combination of structural parameters was obtained by the ASA algorithm. Based on the optimal structural parameter combination, the deformation results are shown in Figure 7, with a model volume of 745.94 mm$^3$ and 84 NSSs filled. According to Equation (12), the displacement rate was calculated to be 3.48%. The optimized prediction results of deformation and displacement rate are compared with the calculated simulation verification results in Table 4. Compared with the initial design model of NSSs with a deformation of 2.85 mm and a displacement rate of 3.63%, the deformation is optimized by 4.6% and the displacement rate is optimized by 4.1%.

**Table 4.** Optimal structure parameter combination and verification results.

| $R_1$ (mm) | $R_2$ (mm) | $R_3$ (mm) | $R_4$ (mm) | Deformation (mm) | | Displacement Rate (%) | |
|---|---|---|---|---|---|---|---|
| | | | | Optimal | Verification | Optimal | Verification |
| 7.89 | 9.09 | 10.17 | 38.42 | 2.75 | 2.72 | 3.48 | 3.48 |

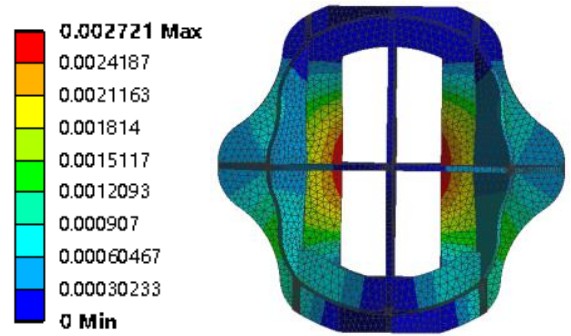

**Figure 7.** Simulation deformation based on ASA.

## 4. The Application of Improved FAHP Method to Multi-Objective Optimization

Some improvements were achieved by the optimization process mentioned above on deformation and displacement rates. However, during the optimization process, the method assumed that the structural parameters ($R_1$, $R_2$, $R_3$, and $R_4$) had equal importance to the objectives (deformation and displacement rate) without allocating weights. The degree of optimization was affected by the results. Therefore, in this work, the MANOVA and an improved FAHP method combined with the ASA algorithm were proposed to calculate weights separately and optimize the objectives (deformation and displacement rate) on the structural parameters ($R_1$, $R_2$, $R_3$, and $R_4$).

As shown in Figures 8 and 9, the contributions of the structural parameter variables $R_1$, $R_2$, $R_3$, and $R_4$ of NSSs were determined by MANOVA on deformation and displacement rate, respectively. From the figures, it can be observed that $R_3$ and $R_2$ have a significant impact on deformation, accounting for 76.69% and 11.47%, respectively. $R_3$, $R_2$, and $R_1$ have a substantial influence on the displacement rate, accounting for 47.95%, 26.48%, and 21.76%,

respectively. Other structural parameters have comparatively smaller effects compared with these variables.

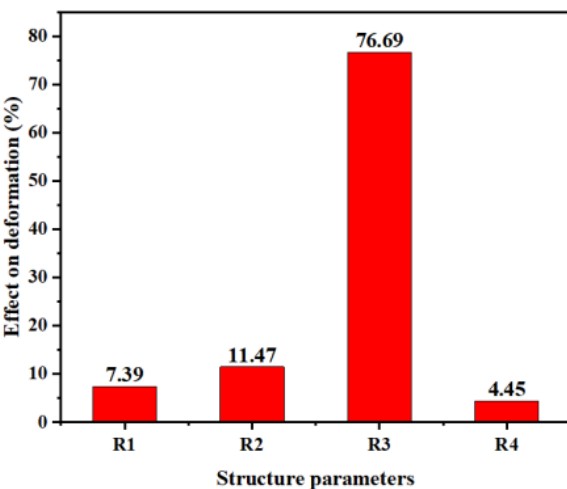

**Figure 8.** Effect of different structure parameters on deformation.

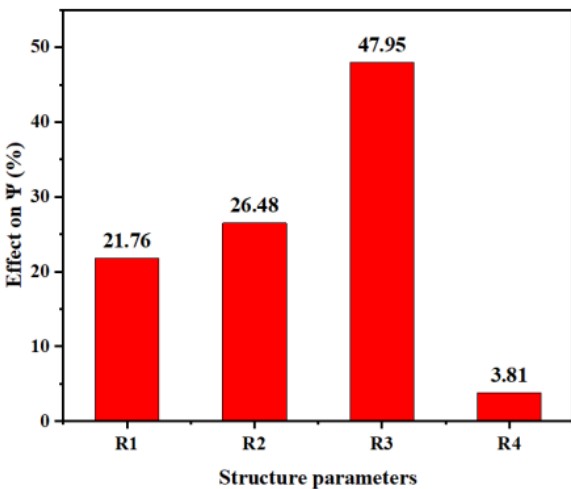

**Figure 9.** Effect of different structure parameters on $\Psi$.

The fuzzy consistency judgment matrix and initial value calculation of weight iteration are performed, and the final weights are computed by iterative calculations of the reciprocal matrix. The results are shown in Tables 5 and 6.

**Table 5.** Weight factors of the different structure parameters on deformation.

| Parameters | $R_1$/mm | $R_2$/mm | $R_3$/mm | $R_4$/mm |
|---|---|---|---|---|
| weight | 0.08 | 0.15 | 0.70 | 0.07 |

**Table 6.** Weight factors of the different structure parameters on $\Psi$.

| Parameters | $R_1$/mm | $R_2$/mm | $R_3$/mm | $R_4$/mm |
|---|---|---|---|---|
| weight | 0.12 | 0.21 | 0.56 | 0.11 |

Taking into account both the deformation and displacement rate objectives, the primary requirement is to satisfy the explosion suppression performance of NSSs, which is the most important indicator. Additionally, a small occupancy space is also desired.

Therefore, there is a slight preference for objectives with minimal deformation compared with objectives with a low displacement rate. According to the improved FAHP method, the weights of $R_1$, $R_2$, $R_3$, and $R_4$ were calculated by Equation (11), and they were 0.096, 0.174, 0.644, and 0.086, respectively. The weight ranking was $R_3 > R_2 > R_1 > R_4$.

Once again, combining the COP prediction model and ASA optimization algorithm, the weights of each factor and objective are allocated to seek the Pareto optimal solution. The maximum number of iterations is set to 200. As shown in Figure 10, the Pareto front is obtained by ASA. Figure 10a–d shows the Pareto front of $R_1$, $R_2$, $R_3$, and $R_4$ on deformation and displacement rate, respectively.

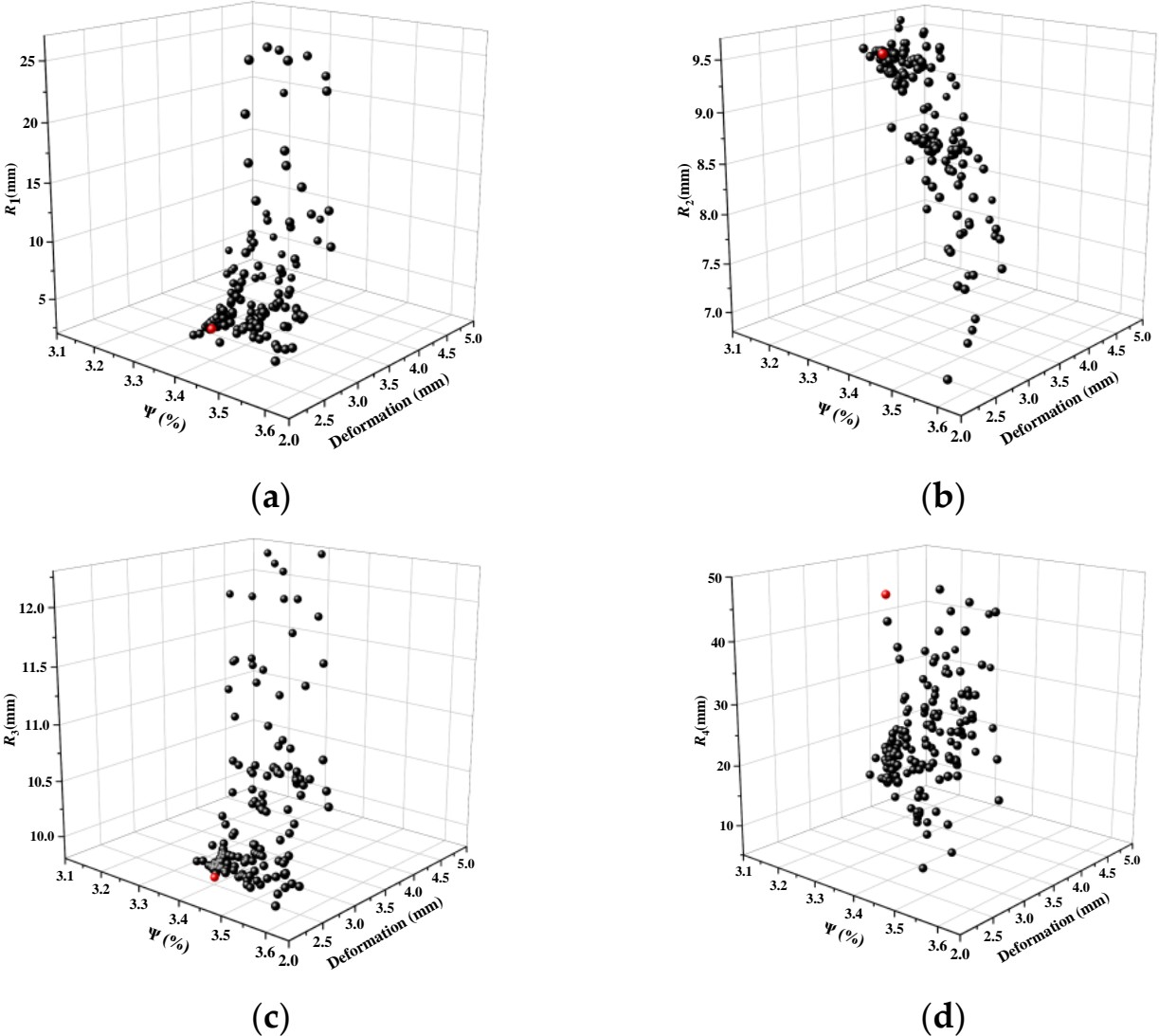

**Figure 10.** The pareto-frontier achieved by ASA and COP models: (**a**) pareto-frontier of $R_1$ on deformation and displacement rate; (**b**) pareto-frontier of $R_2$ on deformation and displacement rate; (**c**) pareto-frontier of $R_{13}$ on deformation and displacement rate; (**d**) pareto-frontier of $R_4$ on deformation and displacement rate.

The red dot in Figure 10 represents the optimal combination of structural parameters in the Pareto front. Red sphere is optimal value. The specific parameters and numerical simulation results are shown in Table 7. The numerical result for deformation is 3.43 mm, and the numerical result for displacement rate is 3.41%.

**Table 7.** Optimal structure parameter combination with weight factor and verification results.

| $R_1$ (mm) | $R_2$ (mm) | $R_3$ (mm) | $R_4$ (mm) | Deformation (mm) | | Displacement Rate (%) | |
|---|---|---|---|---|---|---|---|
| | | | | Optimal | Verification | Optimal | Verification |
| 5.02 | 9.70 | 9.90 | 49.51 | 2.43 | 2.48 | 3.41 | 3.41 |

To validate the accuracy of the optimization results, numerical simulation verification is conducted using displacement rate and statics at the optimal parameters. The model volume of NSSs is 730.73 mm$^3$, with 84 NSSs filled. According to Equation (11), the displacement rate is calculated to be 3.41%. Strength simulation analysis results, as shown in Figure 11, indicate a deformation of 2.48 mm, confirming the correctness of the proposed optimization method. Furthermore, compared with the initial design model of NSSs with a deformation of 2.85 mm and a displacement rate of 3.63%, the deformation had been optimized by 12.98% and the displacement rate had been optimized by 6.1%. Compared with the optimized-design model of NSSs without weight factor with a deformation of 2.75 mm and a displacement rate of 3.48%, the deformation has been optimized by 9.82% and the displacement rate has been optimized by 2.0%. The optimized performance objectives show significant improvement compared with the optimization results without considering weights.

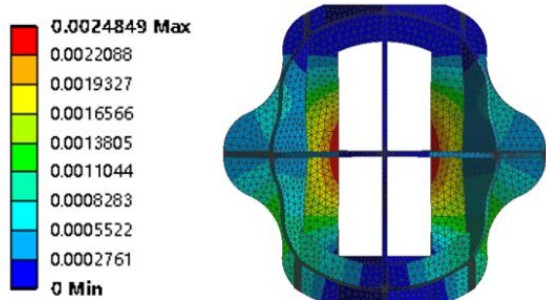

**Figure 11.** The simulation deformation of ASA with weight factor.

## 5. Experimental Verification of NSS Explosion Suppression Performance

Experimental apparatus and procedures are discussed in our work [14]. The equivalence ratio is 1.1. The initial pressure is 0.5 MPa. Weak detonation occurs under these conditions [16]. In our work [14], the filling density of NSSs is 21.9 kg/m$^3$, and its energy attenuation effect is relatively weak. In this work, the same experimental apparatus and procedures are employed to verify the explosion suppression effect of the optimized NSSs. Figure 12a shows the pressure time history of the optimized design NSSs without weight factor and with weight factor conditions. The explosion suppression performance of NSSs was quantitatively analyzed by overpressure contrast. Filtered pressure contrast is analyzed in Figure 12b, which characterizes pressure oscillation contrast and combustion intensity.

As shown in Figure 12, the maximum overpressure of filling the optimized design NSSs without weight factor is 4.07 MPa, and that of filling the optimized design NSSs without weight factor is 3.68 MPa. The maximum overpressure was reduced by 9.58%. Filtered pressure was reduced, and combustion intensity was reduced. As shown in Figure 13, the contrast of flame velocity when it passed through the optimized design NSSs without weight factor and that with weight factor was exhibited. The flame velocity was also significantly reduced. As shown in Figure 14, the contrast of flame propagation when it passed through the optimized design NSSs without weight factor and that with weight factor. The intensity of combustion is reduced. Resulting shock wave is also weakened. Therefore, the suppression effect of NSSs can be enhanced by changing the shape of the NSS spacer reasonably. It was also proved that the energy absorption effect of the spacer on the flame plays an important role in the explosion suppression of NSSs. Weight analysis

was important for the structure of spacers. The multi-objective optimization method with weight analysis was proven.

NSSs are thin-walled skeleton structures. The high surface area ratio of spacers provides a large heat dissipation area to ensure an energy attenuation effect. The mutually connected tiny skeleton structure of NSSs will increase the probability of free radicals hitting spacers and destroying the free radicals, and therefore delay the spreading of flame. The unique skeleton structure can absorb energy to attenuate wavefront pressure through vibration and interfere mechanically. The ability to deform and absorb flame energy is affected by the structure of space. The $R_1$, $R_2$, $R_3$, and $R_4$ are the structures that play a major role in bearing the flame impact and reducing the displacement rate. The optimum condition is reached at 5.02 mm, 9.70 mm, 9.90 mm, and 49.51 mm, respectively.

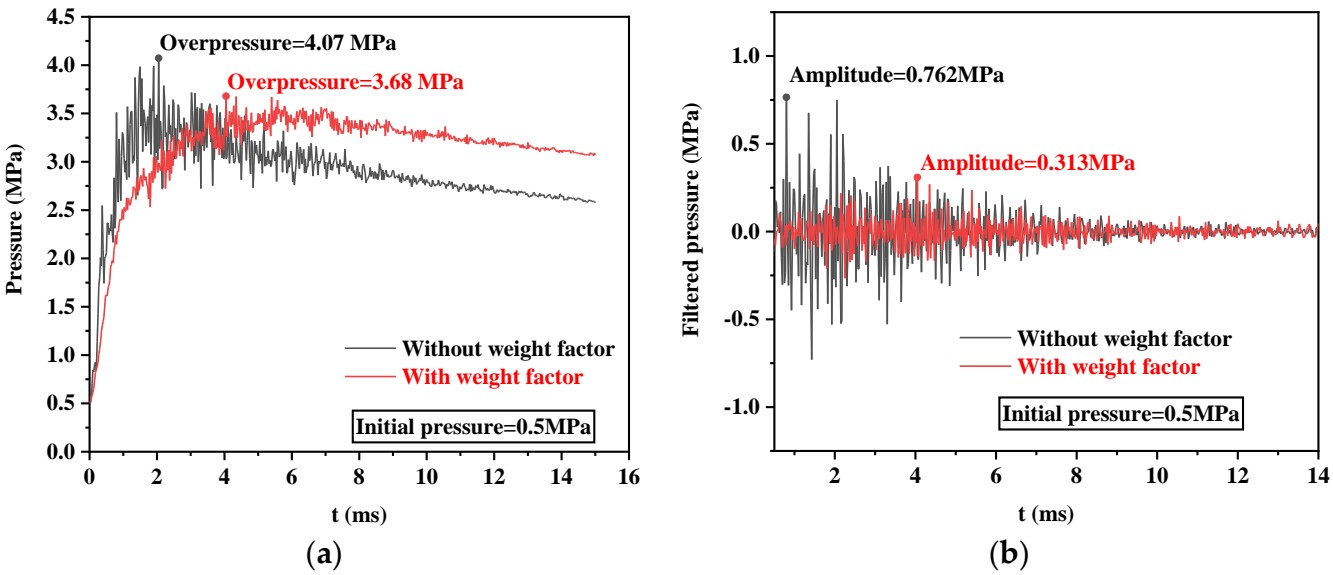

**Figure 12.** Pressure contrast between the optimized-design NSSs without weight factor and those with weight factor. (**a**) overpressure; (**b**) pressure oscillation.

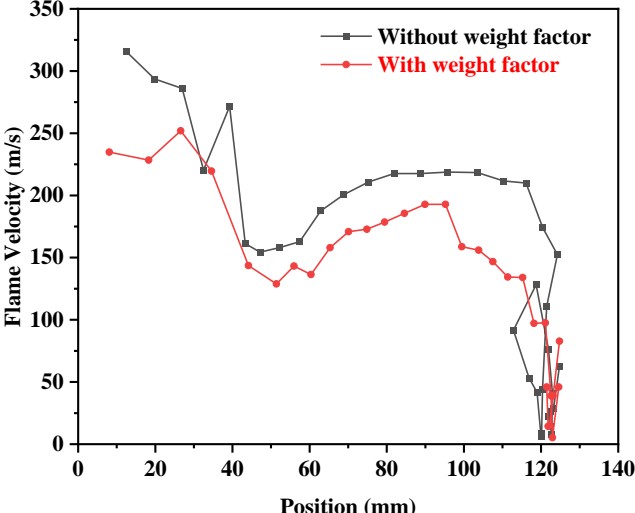

**Figure 13.** Contrast of flame velocity when it passed through the optimized design NSSs without weight factor and that with weight factor.

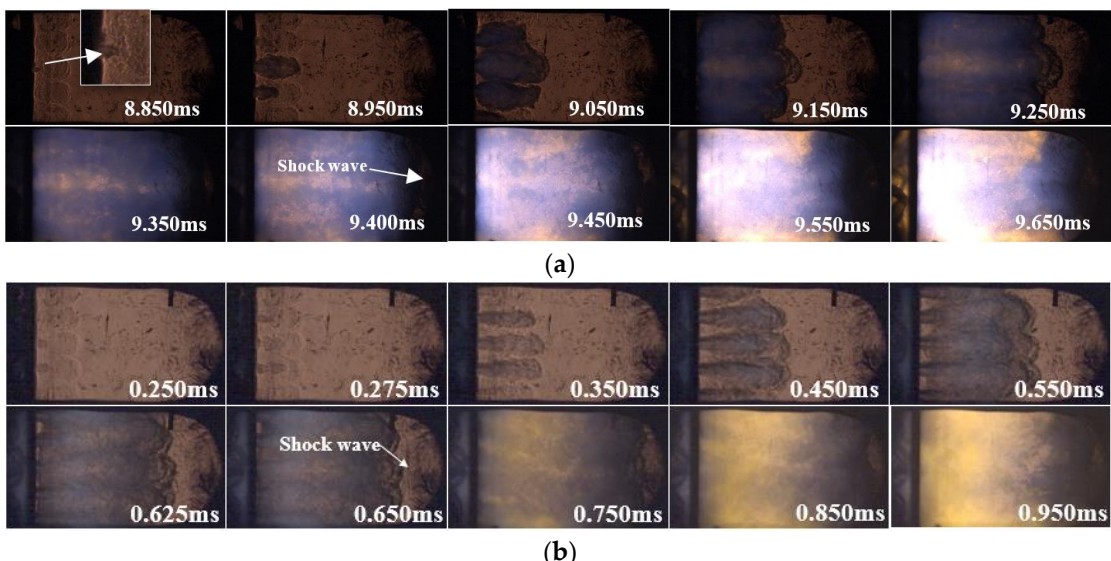

**Figure 14.** Contrast of flame propagation when it passed through the optimized design NSSs without weight factor and that with weight factor. (**a**) without weight factor group; (**b**) with weight factor group.

## 6. Conclusions

(1) The multi-objective optimization method with weight analysis was proposed to optimize the structure of NSSs. The displacement rate and explosion suppression performance were optimized simultaneously. Compared with the original NSS model with a deformation of 2.85 mm and a displacement rate of 3.63%, the deformation is improved by 12.98%, and the displacement rate is improved by 6.1%. Compared with the optimized design model of NSSs without weight factor with a deformation of 2.75 mm and a displacement rate of 3.48%, the deformation has been optimized by 9.82% and the displacement rate has been optimized by 2.0%. The optimized performance objectives show significant improvement compared with the optimization results without considering weights.

(2) One of the multiple objectives was considered a factor; the fuzzy judgment matrix was reconfigured to better reflect actual working conditions, then transformed by membership function conversion. A fuzzy consistency matrix was obtained. Weight allocation was then reassigned, and a multi-objective weight distribution matrix was calculated. The novel FAHP method is proposed to address the weight allocation problem in multi-objective optimization and achieve optimal solutions.

(3) The conflicting multi-objective weight allocation is resolved by MANOVA and the novel FAHP method. Integrating Opt LHD, COP, and ASA, the multi-objective optimization method of structural parameters was developed and applied to the problem of multi-objective optimization of NSSs. The low displacement rate and high explosion suppression performance of NSSs were achieved simultaneously.

(4) This method was developed and applied to the multi-objective optimization of NSSs. At the same time, it was also applied to other structures. But it is not certain whether it can be applied to multi-objective optimization in other areas.

**Author Contributions:** M.L. developed a model of nonmetallic spherical spacers (NSSs) and did experiments on the NSSs; Y.Y. conducted the analyses and wrote the paper; J.Z. contributed experimental equipment and analysis tools; X.Z. developed the actual NSSs; D.W. and M.Y. revised the paper. All authors have read and agreed to the published version of the manuscript.

**Funding:** This research was funded by the Tianjin Science and Technology Program Project, grant number [21ZXCCSN00020].

**Institutional Review Board Statement:** Not applicable.

**Informed Consent Statement:** Not applicable.

**Data Availability Statement:** The data presented in this study are available on request from the corresponding author.

**Conflicts of Interest:** The authors declare no conflict of interest.

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
