# Peer review of "Research on Multi-Objective Optimization on Explosion-Suppression Structure-Nonmetallic Spherical Spacers"

_fire, doi:10.3390/fire7010028_

Round 1

Reviewer 1 Report

Comments and Suggestions for Authors

This is the comments on the Manuscript Number: Manuscript ID: fire-2791513

Type of manuscript: Article
Title:
Research of multi-objective optimization on explosion-suppression structure-Nonmetallic Spherical Spacers based on novel FAHP method with MANOVA

Authors: Minjie Liu , Yangyang Yu 

Rate the Manuscript:

1. Significance to field and specialization of “Fire (ISSN 2571-6255)” journal:     good.

The paper contains the current situation with intense burning phenomena (fire disaster) need to be inhibited in the utilization and transportation process to ensure industrial safety. Nonmetallic Spherical Spacers (NSSs) had been investigated and applied in lots of explosive atmospheres to prevent explosion execution in a confined space. The key structure need to been optimized. In the optimization process, the optimization objective was directly affected by the weight allocation of influencing factors. A novel fuzzy-based analytic hierarchy process (FAHP) is developed to take into account the uncertainty in decision making.

2. Scientific content:   very good.

3. Originality: good.

4. Clarity and presentation:  acceptable.

5. Appropriateness for Journal: appropriate subject matter for the “Fire (ISSN 2571-6255)”.

6. Need for rapid publication: no.

What is the main question addressed by the research?
Do the topic original or relevant to the field? Does it
address a specific gap in the field?

Yes. In this study the optimized target performances were significantly improved compared with the optimization results without weight analysis. It was verified that the suppression effect of NSSs can be enhanced by changing the shape of NSSs spacer reasonably. It was also proved that the energy absorption effect of the spacer on the flame plays an important role in the explosion suppression of NSSs.

Compared with the original NSSs model's deformation (2.85mm) and replacement rate (3.63%), the NSSs with optimized weight allocation had optimized the deformation by 12.98% and replacement rate by 6.1%. Compared to the optimized design model of NSSs without weight factor with deformation of 2.75mm and displacement rate of 3.48%, the deformation has been optimized by 9.82%, and the displacement rate has been optimized by 2.0%.

What specific improvements should the authors consider regarding the
methodology? What further controls should be considered?

OK. In this work, the FAHP method is improved, and the weight matrix Wk 'is calculated to allocate the weight of elements affecting multiple objectives, which effectively solves the problem of factor weight allocation in the multi-objective optimization. Opt LHD, COP and ASA were Combined, a multi-objective optimization method is proposed for structural parameters, high explosion-suppression performance and low replacement rate of NSSs were optimized simultaneously.

Are the conclusions consistent with the evidence and arguments
presented and do they address the main question posed?

Mainly - yes.
Are the
26 references appropriate?

Mainly - yes.
7.
Additional comments on the figures: 1, 10 - can be improved due to instructions to contributors.

Reviewer 2 Report

Comments and Suggestions for Authors

1. The objectives and methodology are to be clear

2.  The abbreviations like Nonmetallic Spherical Spacers (NSSs), fuzzy-based 18 analytic hierarchy process (FAHP) etc. are repeated and to be rectified

3. More references to be reviewed to strengthen the problem-finding part with the following

A] M. Ganesan, K. Balakannan and C. Shankar, (2016), Environmental sustainability evaluation for an automobile manufacturing industry using multi-grade fuzzy approach, International Journal of Engineering Research in Africa, Vol.19, Pp.123-129

4. Table 2 Samples of the structure parameters are to be checked once

5. The future scope and the limitations are to be added

6. The grammatical errors to be checked 

Comments on the Quality of English Language

The grammatical errors to be checked 

Reviewer 3 Report

Comments and Suggestions for Authors

This manuscript reports a multiobjective optimization method for the structural optimization of nonmetallic spherical spacers developed by multifactor analysis of variance and an improved FAHP method that integrates optimal Latin hypercube design, Chebyshev orthogonal polynomials, and adaptive simulated annealing to achieve conflicting high explosion suppression performance and low displacement rates. The logic structure of the manuscript is confusing. In many occasions, the referee cannot understand what the authors want to deliver. The major issues are listed as follows.

1.       The referee wonder what will the authors deliver in the abstract to a reader of interest. For instance, NSSs model's deformation? replacement rate? displacement rate?

2.       Abbreviations should be avoided in the abstract.

3.       The problem formulation should be clearly stated. What are the predictor and the response?

4.       What factors are considered in MANOVA?

5.       The authors should describe the methods in terms of the problem considered.

6.       Please check Eq. (11) and the resulting 3.63%.

7.       Please remove the repetition in Lines 84-97.

8.       The English should be improved significantly.

Comments on the Quality of English Language
English very difficult to understand/incomprehensible.
Extensive editing of English language required.

Reviewer 4 Report

Comments and Suggestions for Authors

Comments to the Author

Authors have submitted the results of a multi-objective optimization on explosion-suppression structure-Nonmetallic Spherical Spacers based on novel 3 FAHP method with MANOVA. Overall, the investigation was performed in a considerable details. But still, the depth of the contents is shallow and the result must be improve. Additionally, some shortcomings are evident as:

-          The title is too long.
The Abstract should be revise to be clearer for the first time reader.

-          In general, the optimization problem is not clearly stated as objective, constraint, design space are either not stated or not visually connected to each other. Please rewrite. Additionally, it is not clear how the presented optimization is actually a multi-objective optimization.

-          Results are presented are noted there is insufficient discussion of the potential reasons for these findings which would be necessary in order for the results to be useful in a wider range of design cases.

-          Finally, the entire manuscript has to be checked carefully for grammatical and terminology errors.

Comments on the Quality of English Language

 the entire manuscript has to be checked carefully for grammatical and terminology errors.

Round 2

Reviewer 4 Report

Comments and Suggestions for Authors

The submission can be accepted as is.

Comments on the Quality of English Language

Still moderate English changes required.